# Silencing the Circadian Clock Genes *Cycle* and *Clock* Disrupts Reproductive–Metabolic Homeostasis but Does Not Induce Reproductive Diapause in *Arma chinensis*

**DOI:** 10.3390/insects16121192

**Published:** 2025-11-23

**Authors:** Junjie Chen, Qiaozhi Luo, Maosen Zhang, Zhuoling Lv, Meng Liu, Xiangchao Huang, Yuyan Li, Lisheng Zhang

**Affiliations:** 1State Key Laboratory for Biology of Plant Diseases and Insect Pests, Key Laboratory of Natural Enemy Insects of Ministry of Agriculture and Rural Affairs, Institute of Plant Protection, Chinese Academy of Agricultural Sciences, No. 2, West Yuan Ming Yuan Road, Beijing 100193, China; 82101211194@caas.cn (J.C.); 15704940113@163.com (Q.L.); zhang_maosen1@163.com (M.Z.); 17769166316@163.com (Z.L.); 18822396348@163.com (M.L.); hxc18058210379@163.com (X.H.); liyuyan@caas.cn (Y.L.); 2College of Horticulture and Landscape Architecture, Tianjin Agricultural University, Tianjin 300392, China; 3Key Laboratory of Animal Biosafety Risk Prevention and Control (North) of Ministry of Agriculture and Rural Affairs, Shanghai Veterinary Research Institute, Chinese Academy of Agricultural Sciences, Shanghai 200241, China

**Keywords:** *Arma chinensis*, circadian clock, RNA interference, reproductive homeostasis

## Abstract

Many insects have an internal biological clock that helps them adapt to daily and seasonal changes. For beneficial insects used in pest control, understanding how this clock works is key to improving their mass production and shelf life. This study focused on two key genes, *Cycle* and *Clock*, that control the internal clock in a predatory insect *Arma chinensis*. We found that these genes are essential for the insect’s normal reproduction and energy balance. When we disrupted these genes, the insects produced fewer eggs, and their energy reserves were disrupted. Importantly, this effect did not trigger diapause, a common dormant state used to survive unfavorable seasons. Our results show that these genes help maintain reproductive and metabolic health under favorable conditions. This knowledge can help develop better methods to rear and store such beneficial insects, supporting more sustainable biological control.

## 1. Introduction

*Arma chinensis* (Hemiptera: Pentatomidae), a generalist predatory bug, preys on over 40 species of agricultural and forestry pests from orders including Lepidoptera and Coleoptera [1,2], making it a highly valuable biological control agent. However, the mass production and commercial application of *A. chinensis* are significantly constrained by its limited shelf life and the challenges of maintaining a constant, high-quality supply. Diapause, a state of programmed developmental arrest and metabolic suppression, offers a potential physiological solution to extend the longevity and preserve the quality of mass-reared natural enemies during storage and transport [3,4]. Consequently, a deeper understanding of the physiological mechanisms regulating dormancy is urgently needed to optimize the application of *A. chinensis*.

Insect diapause is a complex, multifaceted physiological process orchestrated by the precise integration of environmental cues, primarily photoperiod and temperature, with the endocrine system [3,5,6,7,8]. The progression of this complex physiological process is dependent on the precise coordination of internal endocrine signals with external cues, most notably photoperiod and temperature [9]. The circadian clock, an evolutionarily conserved endogenous timekeeping mechanism, is hypothesized to serve as the internal sensor that enables insects to measure photoperiodic changes and transduce this environmental information into downstream hormonal signals [10,11,12,13]. In *Drosophila melanogaster*, the core clock components *Clock* (*Clk*) and *Cycle* (*Cyc*) form a heterodimer that functions as the primary transcriptional activator within the circadian circuitry. This Clk/Cyc complex drives the rhythmic expression of key circadian genes, including *period* (*Per*) and *timeless* (*Tim*), as well as other regulators such as *clockwork orange* (*cwo*), *par-domain protein 1* (*Pdp1*), *vrille* (*vri*), and various *clock-controlled genes* (*ccgs*), thereby generating ~24 h oscillations in physiology and behavior [14,15].

Accumulating evidence underscores the involvement of the circadian clock in the photoperiodic regulation of diapause across a diverse range of insect species. Mutations or knockdowns of core clock genes, including *Tim*, *Cry*, *Per*, *Clk*, and *Cyc*, have been demonstrated to disrupt photoperiodic diapause induction in taxa such as *Chymomyza costata* [16], *Bombyx mori* [17,18,19], *Helicoverpa armigera* [20], *Pyrrhocoris apterus* [21,22], et al. While these studies traditionally focus on the clock’s role in seasonal adaptation, a fundamental yet often overlooked aspect is its dual function in maintaining daily metabolic and reproductive homeostasis. Beyond its role as a seasonal timer, the circadian clock is integral to the daily coordination of energy metabolism and reproductive processes, ensuring resources are allocated efficiently for ovariogenesis and vitellogenesis under favorable conditions [23]. *A. chinensis* enters reproductive diapause under short-day (LD 8:16) and low-temperature conditions, with the photoperiod-sensitive stage occurring within the first 24 h after adult emergence [24,25,26], making it an excellent model for investigating the initial molecular events of photoperiodic response. While the low titers of juvenile hormone (JH) have been established as a key endocrine feature of its reproductive diapause [27], the upstream regulatory mechanisms that link photoperiod perception to JH signaling and the maintenance of reproductive–metabolic homeostasis remain largely unexplored. Although *Clk* and *Cyc* are recognized as central positive regulators of the circadian clock and are crucial for diapause in other insects, their specific roles in predatory natural enemies, particularly in *A. chinensis*, are unknown.

In this study, we first cloned and characterized *AcClk* and *AcCyc* from *A. chinensis* and analyzed their spatiotemporal and diel expression patterns. We then employed RNA interference (RNAi) to investigate the functional consequences of *AcClk* and *AcCyc* knockdown on ovarian development, fecundity, and energy metabolism under non-diapause conditions. Finally, we explored the regulatory relationship between these clock genes and the JH signaling pathway. Our findings showed that the core circadian clock genes *Clk* and *Cyc* are essential maintainers of reproductive and metabolic homeostasis but not merely regulators of diapause induction under favorable conditions, facilitating the development of strategies for improving the mass production and shelf life of this economically important biological control agent.

## 2. Materials and Methods

### 2.1. Insect Rearing and Sample Preparation

The tested *A. chinensis* populations were collected from Langfang Base Laboratory in Hebei Province, Institute of Plant Protection, Chinese Academy of Agricultural Sciences. Silkworm pupae were used to rear *A. chinensis* for successive generations to form a stable population for subsequent experiments. Non-diapausing rearing conditions for *A. chinensis*, 26 °C, with a long-day photoperiod of 16 h light: 8 h dark per 24 h, and a relative humidity (RH) of 70 ± 5%. The diapause induction conditions for *A. chinensis* were as follows: 15 °C/5 °C, with a short-day photoperiod of 8 h light: 16 h dark and a relative humidity RH of 70 ± 5% [24]. Our previous studies have shown that *A. chinensis* has a sensitive period within the first 24 h of initial emergence, during which it is particularly prone to entering diapause [24]. For analyzing temporal expression profile of *AcCyc* and *AcClk* genes, samples included female adults at 0, 3, 6, 9, 12 and 15 days under non-diapause conditions, and at 0, 10, 20, 30, 40, and 50 days under diapause conditions. Five tissues (head, ovary, fat body, midgut, and malpighian tubule) were collected. All samples were collected and stored at −80 °C until analysis was performed. All samples contained three biological replicates to RT-qPCR analysis with each biological replicate consisting of five females.

### 2.2. Molecular Cloning and Sequence Analysis

The sequences of *AcCyc* and *AcClk* were obtained from previous laboratory transcriptome data. Sequence fragments of *AcCyc* and *AcClk* were amplified using specific primers (Appendix A) and cDNA, ligated into the pMD18-T plasmid (Takara Bio, Otsu, Japan), sent to Tsingke Bio (Beijing, China) for sequencing, and then assembled to obtain the ORF sequences. Homologous sequences of *AcCyc* and *AcClk* from other insects were obtained from the National Center for Biotechnology Information (NCBI) (Appendix A) and analyzed using DNAman 6.0.3 software. ClustalW 2 and ESPcript 3.0 Web were used for sequence analysis. The phylogenetic analysis of *AcCyc* and *AcClk* was conducted using the Jones–Taylor–Thornton (JTT)-based neighbor-joining method with 1000 bootstrap replicates in MEGA 7.0 software.

### 2.3. Real-Time Fluorescence Quantitative PCR Analysis

Total RNA was extracted from the insect samples using the TRIzol Reagent, and 1 μg of total RNA was reverse-transcribed to cDNA using TransScript^®^ One-Step gDNA Removal and cDNA Synthesis SuperMix (TransGen Biotech, Beijing, China) under the following conditions: 42 °C for 30 min, and 85 °C for 5 s. Subsequently, real-time PCR was performed using the TOROGreen^®^ 5G qPCRPremix (Toroid Technology Limited, Norwich, UK) and a LightCycler^®^ 96 Instrument (Roche, Basel, Switzerland). All reactions were run in triplicates, with a total volume of 20 μL, and each reaction consisted of 10 μL TOROGreen Premix, 0.8 μL of each specific primer (Appendix A), 1 μL sample cDNA, and 7.4 μL nuclease-free water. PCR amplification of the genes was conducted under the following conditions: 95 °C for 5 min, followed by 40 cycles at 98 °C for 10 s and 55–59 °C for 20 s. A dissociation step cycle (95 °C for 10 s, 65 °C for 60 s, and from 65 °C to 97 °C in increments of 0.2 °C/s; 5 readings/°C) was added for the melting curve analysis. When the reactions were complete, CT values were determined using fixed threshold settings. The primers used for RT-qPCR are listed in Appendix A, and *RPL27* was used as endogenous reference gene [25,26]. The 2^−ΔΔCT^ method was used to analyze the relative expression levels of genes [28].

### 2.4. Diel Expression of AcCyc and AcClk Under Non-Diapause or Diapause Conditions

To elucidate the effects of non-diapause and diapause conditions on the circadian rhythm expression of *AcCyc* and *AcClk*, female adults that had emerged for 20 days under both conditions were sampled every 3 h. Each sample consisted of three biological replicates, with each replicate containing eight adult heads. Samples collected at Zeitgeber Time ZT 3, ZT 6, ZT 9, ZT 12, ZT 15, ZT 18, ZT 21 and ZT 24. All samples were collected for RNA extraction, cDNA synthesis and RT-qPCR detection.

### 2.5. RNA Interference Bioassays

To determine the functions of *AcCyc* and *AcClk*, RNA interference (RNAi) was used to knock down gene expression. The 372 bp region of dsAcCyc and the 760 bp region of dsAcClk were amplified from cDNA using specific primers (Appendix A). Double-stranded RNAs (dsRNAs) were then synthesized using the MEGAscript T7 High Yield Transcription Kit (Invitrogen, Carlsbad, CA, USA) from amplified target fragments. These fragments were amplified using specific primers containing the T7 promoter sequence (Appendix A) and cDNA. Using a Nanoject III microsyringe (Drummond Scientific Company, Broomall, PA, USA), 1 μL (2 µg/μL) of dsRNA was injected into the second interstitial membrane of the abdomen in female adults 24 h after emergence. Samples were collected at 24, 48, and 72 h post-injection to assess the efficiency of *AcCyc* and *AcClk* knockdown. The expressions of *AcPer* gene, *AcTim* gene, and juvenile hormone pathway genes, *juvenile hormone acid methyltransferase enzymes* (*AcJHAMT1*, *AcJHAMT2*, *AcJHAMT3*), *AcKr-h1* and *AcMet* were assessed at 48 h post-injection. Adults were injected with dsGFP as a negative control. Each sample consisted of three biological replicates, with each replicate containing eight adults.

To examine the effects of *AcCyc* and *AcClk* knockdown on ovarian development under non-diapause conditions, newly emerged female adults were injected with dsAcCyc, dsAcClk, and dsGFP, respectively, then maintained under non-diapause conditions (26 °C under a 16 L:8 D photoperiod) after which the ovaries were dissected and isolated. During this period, equal amounts of dsAcCyc, dsAcClk, and dsGFP were injected into female adult insects treated with the corresponding dsRNA every 4 days. Ovarian samples were photographed and the length and width were measured using a Leica VHX-2000 stereomicroscope (Keyence (China) Co., Ltd., Beijing, China). Meanwhile, female adults treated as described above were collected for RNA extraction and cDNA synthesis to detect the relative expression level of the *AcVg* by qPCR. Each sample consisted of three biological replicates, with each replicate containing eight adults.

In the egg-laying experiment, after *GFP*, *AcCyc*, and *AcClk* dsRNA injection, females were paired with males and raised under non-diapause conditions, with the number of eggs laid within 30 days recorded. The injection method and frequency of dsRNA were consistent as mentioned above. Additionally, female adults of uniform size were weighed before injection and again 8 days later.

To assess the effect of gene knockdown on energy metabolism, female adults injected with *AcCyc* and *AcClk* dsRNA were analyzed total lipid after 72 h. After injecting of 1 μL of dsRNA (dsAcCyc, dsAcClk, dsGFP) into *A. chinensis* females, fresh food was provided daily under their respective feeding conditions. The determination method of total lipid content refers to previous studies [29]. Each treatment consisted of eleven biological replicates, with each replicate containing three adults.

### 2.6. The Assessement of Potential Off-Targets of dsAcCyc and dsAcClk

For reasonably identify the downstream genes of *AcCyc* and *AcClk*, the potential off-targets of dsAcCyc and dsAcClk in *C. septempunctata* were evaluated using “dsRIP” (https://dsrip.uni-goettingen.de/efficiency) (accessed on 10 November 2025) with 1 mismatche per siRNA for off-target prediction.

### 2.7. Statistical Analysis

The experimental data were analyzed, and Graphs were created using GraphPad Prism 9.0 (GraphPad Software, San Diego, CA, USA). Data were expressed as the mean ± standard error (SE) of three independent replicates. The expression profiles of *AcCyc* and *AcClk* at different developmental stages or in different tissues were analyzed by one-way ANOVA, followed by a Turkey’s HSD multiple comparison test with a, b and c indicating signification differences at *p* < 0.05. The diel expression of *AcCyc* and *AcClk* under non-diapause and diapause conditions were analyzed by one-way ANOVA, followed by a Turkey’s HSD multiple comparison test, with a, b and c indicating signification differences at *p* < 0.05. The knockdown efficiency, TAG content, ovary sizes (length and width) and expression of genes following *AcCyc* or *AcClk* silencing were analyzed by Student’s *t*-test, and significance levels were denoted by * (0.01 ≤ *p* < 0.05), ** (0.001 ≤ *p* < 0.01), *** (*p* < 0.001) and **** (*p* < 0.0001).

## 3. Results

### 3.1. Identification and Characterization of the A. chinensis AcCyc and AcClk Gene

The open reading frame (ORF) of *AcCyc and AcClk* were 849 bp and 1725 bp, encoding 282 amino acids and 574 amino acids, respectively. The phylogenetic trees were constructed to investigate the phylogenetic relationships of various Cyc and Clk protein. The results revealed that both AcClk and AcCyc (Figure 1) were clustered within clades comprising their respective orthologs from Hemiptera insects, showing particularly close relationships to proteins from *Halyomorpha halys* Clock and *Halyomorpha halys* Cycle.

### 3.2. Spatiotemporal Expression Patterns of AcCyc and AcClk During Diapause and Non-Diapause Conditions

RT-qPCR results showed the mRNA abundance of *AcCyc* gene was the highest in newly emerged females (NE) at diapause or non-diapause conditions (Figure 2A,B). The spatial expression profile showed that *AcCyc* gene was highest expressed in the midgut at diapause or non-diapause conditions (Figure 2C,D). In the non-diapause condition, the expression of *AcClk* was the highest in NE and N3 and the lowest in N15 (Figure 2E). Under diapause conditions, the highest expression of *AcClk* was in D50 (Figure 2F). The mRNA abundance of *AcClk* was higher in the head and fat body and was lowest in the midgut under non-diapausing conditions (Figure 2G). Under diapause conditions, the mRNA abundance of *AcClk* was highest in the head and was lowest in the ovary and fatbody (Figure 2H). These results suggested that *AcCyc* and *AcClk* may be involved in the non-diapause induction process of *A. chinensis*.

### 3.3. Rhythmic Expression of AcCyc and AcClk in Adults During Diapause and Non-Diapause Conditions

Diel expression patterns of *AcCyc* and *AcClk* were examined after 20 days of non-diapause and diapause conditions. The results showed that in the non-diapause 20-day-old, the *AcCyc* gene was up-regulated at ZT6 and ZT18, and the expression of *AcCyc* was continuously downregulated from ZT6 to ZT15 in the bright period (Figure 3A). At 20 days of diapause, the mRNA abundance of *AcCyc* gene was relatively stable from ZT3 to ZT9 in the bright period (Figure 3B). The expression of *AcCyc* was up-regulated from ZT9 to ZT24 in the dark period (Figure 3B). In the non-diapause 20-day-old, The *AcClk* gene was at lower expression level from ZT3 to ZT6 in the bright period (Figure 3C). The expression level of *Clk* genes showed higher expression level at ZT9 and ZT12 in the bright period and at ZT18 in the dark period (Figure 3C). In the 20th day of diapause, *AcClk* gene was lower expressed in the bright period, while was higher expressed in the dark period (Figure 3D), Therefore, the expression pattern of circadian clock genes in non-diapause and diapause differences suggested the key function of circadian clock genes in *A. chinensis*.

### 3.4. Silencing AcCyc and AcClk Genes Impeded Ovarian Development and Reduced Triglyceride (TAG) Content in Non-Diapausing Females

To investigate the functional roles of the circadian clock genes *AcCyc* and *AcClk* in reproductive diapause, RNA interference (RNAi) was employed to knock down these genes in non-diapausing adult females of *Arma chinensis*. Knockdown efficiency was robust and time-dependent. For *AcCyc*, interference efficiencies reached 84%, 84%, and 83% at 24 h, 48 h, and 72 h post-injection of dsRNA, respectively (Figure 4B). Similarly, silencing of *AcClk* resulted in efficiencies of 90%, 76%, and 73% at the corresponding time points (Figure 4D). Phenotypic observations revealed substantial reproductive impairments. Although vitellogenin deposition was detectable indsAcCyc, dsAcClk and dsGFP control groups, it was markedly reduced following RNAi (Figure 4A,C). Ovarian development was significantly compromised, with reduced ovarian length and width in the *AcCyc*-silenced group (Figure 4E,F), while only ovarian width was significantly reduced upon *AcClk* knockdown (Figure 4H,I). Consistent with these morphological defects, transcriptional analysis showed that *vitellogenin (Vg)* expression was significantly suppressed after silencing of either gene, with a 75% reduction observed in *AcCyc*-knockdown females (Figure 4G) and a 77% reduction observed in *AcClk*-knockdown females (Figure 4J). Furthermore, reproductive performance was severely affected. Silencing of *AcClk* delayed oviposition onset until the 18th day and significantly reduced total egg production (Figure 5A). Similarly, *AcCyc* knockdown shortened the pre-oviposition period by 3 days and also led to a significant decrease in fecundity (Figure 5A). These results indicate that both *AcCyc* and *AcClk* are critical for normal ovarian development and timely oviposition, and their disruption leads to declined reproductive capacity in *A. chinensis* under non-diapause conditions. In addition, compared with the control group, the TAG content was reduced by 44.74% after *AcCyc* (Figure 5B) knockdown and by 29.82% after *AcClk* silencing (Figure 5C).

### 3.5. Silencing AcCyc and AcClk Genes Regulate Other Circadian Clock Genes

Compared with the dsGFP control group, the expressions of *AcPer (*Figure 6A), *AcTim* (Figure 6B) and *AcClk* (Figure 6C) were significantly reduced after injection of dsAcCyc in non-diapause female adults. Additionally, knocking down the *AcClk* downregulated the expression levels of *AcPer (*Figure 6E) and *AcCyc* (Figure 6G) but led to no significant change in downregulated *AcTim* (Figure 6F).

### 3.6. Silencing AcCyc and AcClk Genes Downregulate Juvenile Hormone Pathway Genes

Compared with the dsGFP control group, the expression of *AcKr-h1 (*Figure 6D) and *AcMet* (Figure 6L) was significantly reduced after injection of dsAcCyc in non-diapause female adults, but the expression levels of *JHAMT1*, *JHAMT2*, and *AcJHAMT3* were not significantly affected (Figure 6I–K). Additionally, knocking down the *AcClk* led to no significant change in the expression levels of *JHAMT1*, *JHAMT2*, and *AcJHAMT3* (Figure 6M–O) but also downregulated *AcKr-h1* (Figure 6H) and *AcMet* (Figure 6P).

### 3.7. Potential Off-Targets of dsAcCyc and dsAcClk

The analysis revealed that dsAcClk was designed to only specifically target *AcClk* gene (Cluster-9636.42629: *A. chinensis* Clock), and no potential off-target effects were detected, theoretically (Appendix A; Appendix A). The dsAcCyc was designed to specifically target *AcCyc* gene (Cluster-9636.39048: *A. chinensis Cycle*) and two potential off-target genes (Cluster-9636.34927 and Cluster-9636.74299: both *A. chinensis* uncharacterized genes), theoretically (Appendix A; Appendix A).

## 4. Discussion

Our results provide compelling evidence that in the predatory bug *A. chinensis*, *AcClk* and *AcCyc* are indispensable regulators of reproductive and metabolic homeostasis under favorable (non-diapause) conditions. Their functional disruption leads to a suite of physiological deficits, impaired ovarian development, suppressed vitellogenin expression, reduced fecundity, and dysregulated energy storage, that collectively resemble, but are fundamentally distinct from, the diapause syndrome.

The robust spatiotemporal expression rhythms of *AcClk* and *AcCyc* under both diapause and non-diapause conditions, and their significant differential expression between these states, underscore their central role in photoperiodic time measurement. Crucially, RNAi-mediated silencing did not induce a typical, adaptive reproductive diapause, which is characterized by programmed ovarian arrest and strategic triglyceride (TAG) accumulation for survival. Instead, it precipitated a pathological state of dysfunction. The observed decline in TAG content, coupled with reproductive failure, indicates a collapse of physiological systems rather than an induction of a dormant, energy-conserving state. The observed reproductive failure likely contributes by decline in TAG content, as sufficient energy reserves are essential to fuel the energetically costly processes of vitellogenesis and oogenesis in insect [30,31,32]. These suggests that the primary role of these genes under favorable conditions is to maintain the integrity of reproductive and metabolic processes, ensuring resources are efficiently allocated for vitellogenesis and oogenesis. The phenotypic evidence for this reproductive impairment is clear. RNAi knockdown resulted in severely compromised ovarian development, as evidenced by significant reductions in ovarian size. This was coupled with a dramatic suppression of *Vg* gene expression and a profound decline in fecundity. Our findings provide compelling evidence that in the predatory *A. chinensis*, *AcClk* and *AcCyc* function as indispensable regulators of reproductive–metabolic homeostasis under favorable (non-diapause) conditions. This aligns with observations in other hemipteran species. For instance, in the brown-winged green stink bug *Plautia stali*, RNAi-mediated knockdown of *Cyc* disrupts the photoperiodic response, leading to suppressed ovarian development [33]. Specifically, *Cyc* RNAi resulted in inhibited ovarian development in 45% of individuals under long-day conditions, a phenotypic outcome consistent with the effects observed following *AcCyc* knockdown in our study. These collective results suggest a conserved role for the *Cycle* in Hemiptera, underscoring its core function in the daily coordination of reproduction.

The molecular mechanism underlying this pleiotropic phenotype appears to be the specific disruption of downstream signaling within the juvenile hormone (JH) pathway, rather than an alteration in JH biosynthetic capacity. Our data clearly demonstrate that knockdown of *AcCyc* or *AcClk* led to the significant downregulation of critical JH pathway components, including the receptor *AcMet* and the responsive transcription factor *AcKr-h1*. The dramatic reduction in *Met* expression following *AcCyc* and *AcClk* knockdown implied genetic interactions between circadian clock components and JH signaling pathways. This genetic interaction between circadian clock components and JH signaling elements (eg, Met, Taiman), also previously reported in photoperiodic regulation of *P. apterus* [34,35], appears to be a conserved regulatory module in Hemiptera insects. *Juvenile hormone acid methyltransferase enzyme* (*JHAMT*), a gene encoding a key rate-limiting enzyme in the juvenile hormone (JH) biosynthetic pathway, has been extensively studied. Most experimental evidence indicates that knockout of the *JHAMT* gene leads to a decrease in JH titer within insects, consequently inducing reproductive diapause [36,37,38,39]. However, the expression levels of three *JHAMT* genes, *AcJHAMT1*, *AcJHAMT2*, and *AcJHAMT3*, remained unchanged following RNAi. The copy number of *JHAMT* genes exhibits considerable variation across arthropod species. While some insects, such as *Drosophila melanogaster*, *Apis mellifera*, and *Aphis craccivora*, possess only a single *JHAMT* gene in their genomes [40], the occurrence of multiple *JHAMT* paralogs is remarkably widespread in insects, including in the desert locust *Schistocerca gregaria* (31 *JHAMT-like* genes), the housefly *Musca domestica* (12 *JHAMT* genes), and the silkworm *Bombyx mori* (6 *JHAMT* genes) [40]. However, the presence of multiple *JHAMT* gene copies does not imply that all are functionally involved in JH biosynthesis. For example, in the red flour beetle, *Tribolium castaneum*, which possesses three *JHAMT* genes, only one encodes an enzyme with genuine JH acid methyltransferase activity, while the other two paralogs lack this catalytic function [41]. It is important to note that, although direct evidence is currently lacking to confirm that all three *AcJHAMT* genes are functionally involved in JH synthesis in *A. chinensis*, the absence of significant changes in their transcript levels following RNAi-*AcCyc*/*AcClk* may suggest that the transcriptional regulation of JH biosynthesis was not substantially impacted after RNAi-*AcCyc*/*AcClk*. Therefore, we propose that *AcClk* and *AcCyc* act as upstream regulators that maintain the JH pathway in an active state under non-diapause conditions, thereby ensuring the continuity of reproductive processes.

Furthermore, recent research in Lepidoptera has revealed that the circadian clock gene *Cycle* exhibits functional polymorphism through alternative isoforms, which play distinct roles in diapause regulation [42]. In the silk moth *Bombyx mori*, Zheng et al. (2025) demonstrated that Cyc encodes three major isoforms: *CycA* and *CycB* are primarily involved in core circadian rhythm maintenance, while *CycC* specifically regulates diapause entry [42]. A critical deletion in the *CycC* isoform disrupts diapause induction in polyvoltine strains, leading to non-diapause phenotypes, without affecting circadian rhythms controlled by *CycA/B*. This isoform-specific function is conserved across Lepidoptera, including distantly related species like the Asian corn borer (*Ostrinia furnacalis*), where *CycC* knockdown reduces diapause incidence [42]. These findings provide a nuanced perspective on our results in *A. chinensis*, where knockdown of *AcCyc* under non-diapause conditions impaired reproductive–metabolic homeostasis (e.g., reduced ovarian development, JH signaling disruption) but did not induce typical diapause. This suggests that in *A. chinensis*, the *Cyc* ortholog may function analogously to the rhythm-regulating isoforms (*CycA/B*) rather than the diapause-specific *CycC*, potentially due to evolutionary divergence in Hemiptera. Notably, our results are different from recent findings in other Coleoptera [43]. Gao et al. (2025) demonstrated that in the ladybird beetle *Harmonia axyridis*, *Clk* and *Cyc* regulate winter diapause entry through a noncanonical pathway involving the NuA4/TIP60 histone acetyltransferase complex [43]. In *H. axyridis*, knockdown of *Clk* and *Cyc* under long-day conditions induced diapause-like phenotypes via disruption of juvenile hormone (JH) biosynthesis, whereas knockdown of *Per* or *Tim* had no effect, a dissociation similar to what we observe in *A. chinensis*. This comparison underscores that circadian clock genes can function through rhythm-independent mechanisms across insects, but their specific outcomes (e.g., diapause induction vs. homeostasis maintenance) may be species- or context-dependent.

The interpretation of our RNAi results is complicated by the interconnected nature of the circadian clock network. The dsRNA-mediated knockdown of *AcCyc* non-specifically suppressed the expression of other core clock genes, including *AcPer*, *AcTim*, and *AcClk*. Similarly, knocking down *AcClk* also suppressed the expression of *AcPer* and *AcCyc*. This suggests a potential mutual positive feedback regulation within the clockwork, a phenomenon observed in other insects where clock genes are inter-regulated to maintain the ~24 h oscillation. Therefore, the observed phenotypic effects may result from a partial disruption of the entire clock circuit rather than from the loss of a single gene’s function alone [44,45].

The absence of a complete and immediate diapause phenotype following RNAi could be attributed to several factors: (1) functional redundancy or compensatory mechanisms within the circadian clock network, where other genes buffer the loss of a single component; (2) the action of downstream outputs that remain partially active even when the core clock is dampened; or (3) species-specific differences in the reliance on clock genes for diapause induction. Furthermore, RNAi knockdowns are inherently imperfect. The residual mRNA and protein activity might be sufficient to prevent the full manifestation of the phenotype, a possibility underscored by the delayed egg-laying observed in dsAcClk-treated insects (Figure 5), which suggests a partial rather than absolute loss of function. While our RNAi efficiencies were high for both dsAcCyc and dsAcClk (Figure 4B,D), we acknowledge the inherent limitations of the technique, including the potential for non-specific off-target effects (theoretically predicted for dsAcCyc in Appendix A and Appendix A). The RNAi approach, while powerful, can lead to non-specific effects and typically results in partial rather than complete loss-of-function phenotypes, potentially underestimating a gene’s role if compensatory mechanisms exist. Future investigations employing more precise gene-editing technologies like CRISPR/Cas9, or combinatorial gene knockdowns, would provide deeper insights into the functional hierarchy within the clock and its outputs.

## 5. Conclusions

In conclusion, we reframe the role of *AcClk* and *AcCyc* from being mere potential triggers for diapause to being essential guardians of reproductive–metabolic homeostasis. Their dysfunction disrupts JH signaling, leading to a breakdown in reproduction and energy management. From an applied perspective, this study offers valuable insights for improving the mass production of *A. chinensis*. Understanding that these clock genes are vital for maintaining robust reproductive output suggests that optimizing rearing conditions (e.g., light cycles) to support their natural rhythmic expression could enhance the quality and fecundity of mass-reared populations, thereby supporting more effective biological control programs.

## Figures and Tables

**Figure 1 insects-16-01192-f001:**
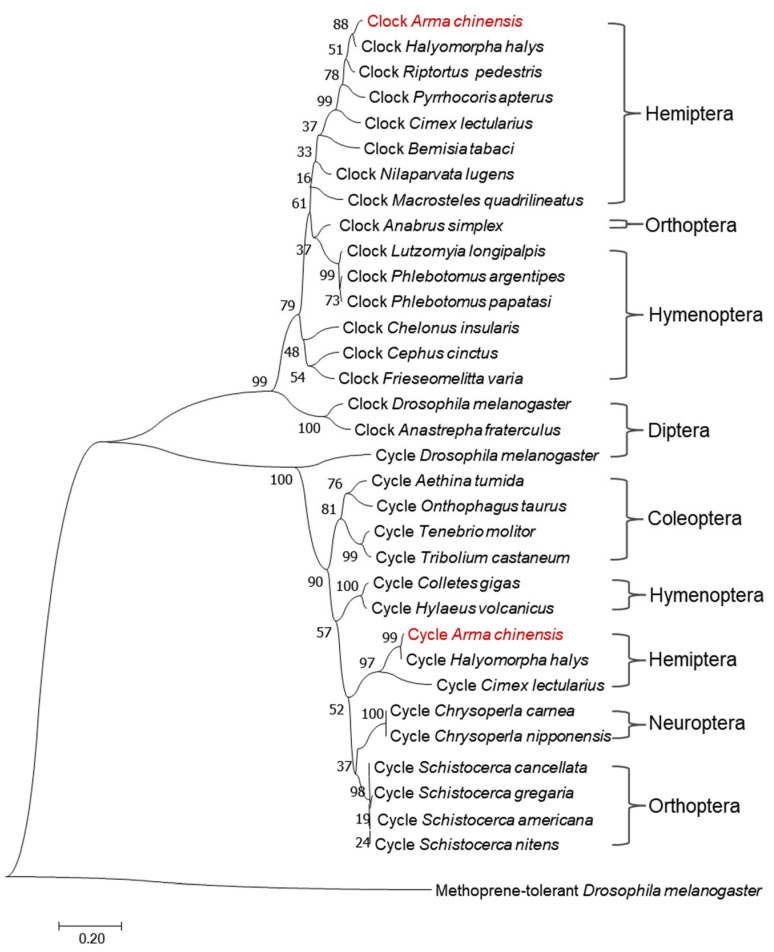
Evolutionary relationship of Cycle and Clock protein sequences. The phylogenetic trees were constructed based on amino acid sequences of Cycle and Clock using the Jones–Taylor–Thornton (JTT)-based neighbor-joining method with 1000 bootstraps.

**Figure 2 insects-16-01192-f002:**
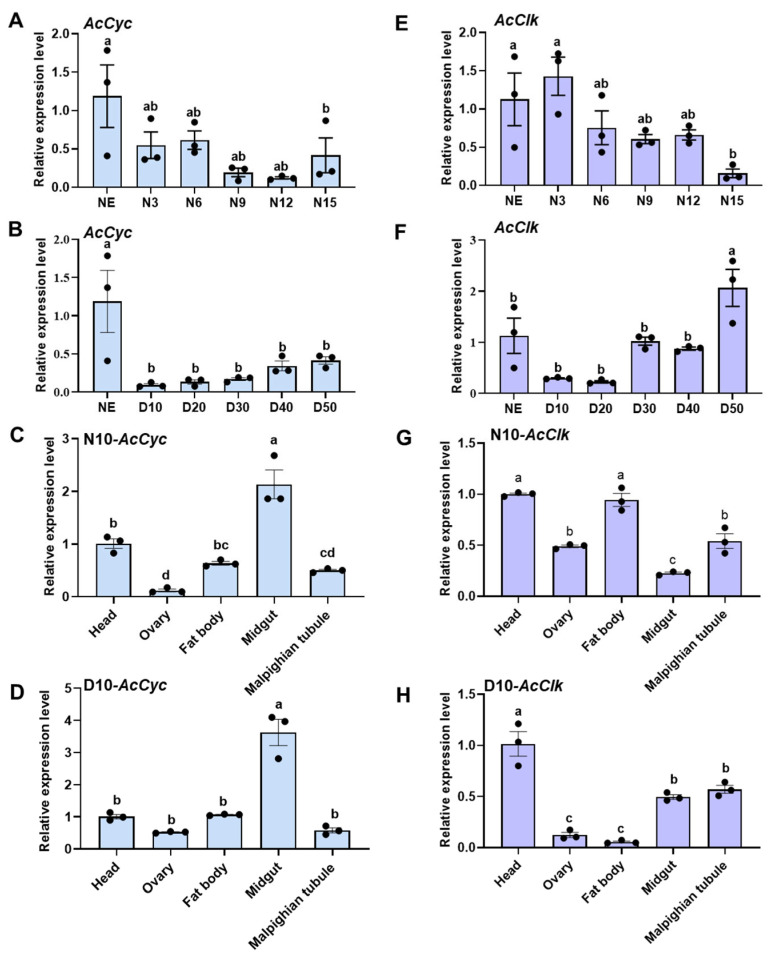
Expression profiles of *AcCyc* and *AcClk* at different developmental stages and in different tissues. Temporal expression patterns of *AcCyc* and *AcClk* under non-diapause (**A**,**E**) and diapause conditions (**B**,**F**). Spatial transcript expression analysis of *AcCyc* and *AcClk* under non-diapause (**C**,**G**) and diapause conditions (**D**,**H**). Data represent mean ± stand error of mean (SEM). Different letters indicated statistically significant differences between different samples using a one-way analysis of variance (ANOVA) with Tukey’s multiple comparisons test, *p* < 0.05. NE, N3, N6, N9, N12 and N15, respectively, represented the female adults at 0, 3, 6, 9, 12, and 15 days under non-diapause conditions. NE, D10, D20, D30, D40 and D50, respectively, represented the female adults at 0, 10, 20, 30, 40, and 50 days under diapause conditions.

**Figure 3 insects-16-01192-f003:**
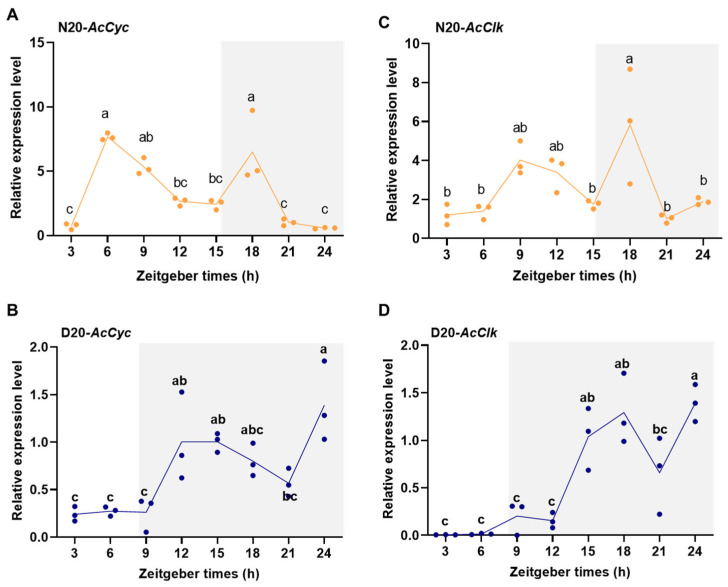
Diel expression of *AcCyc* and *AcClk* under non-diapause and diapause conditions. (**A**,**C**) Circadian rhythm expression pattern of *AcCyc* and *AcClk* under non-diapause condition. (**B**,**D**) Circadian rhythm expression pattern of *AcCyc* and *AcClk* under diapause condition. The gray shadow represented dark period. Data represent mean ± stand error of mean (SEM). Different letters indicated statistically significant differences using a one-way analysis of variance (ANOVA) with Tukey’s multiple comparisons test, *p* < 0.05.

**Figure 4 insects-16-01192-f004:**
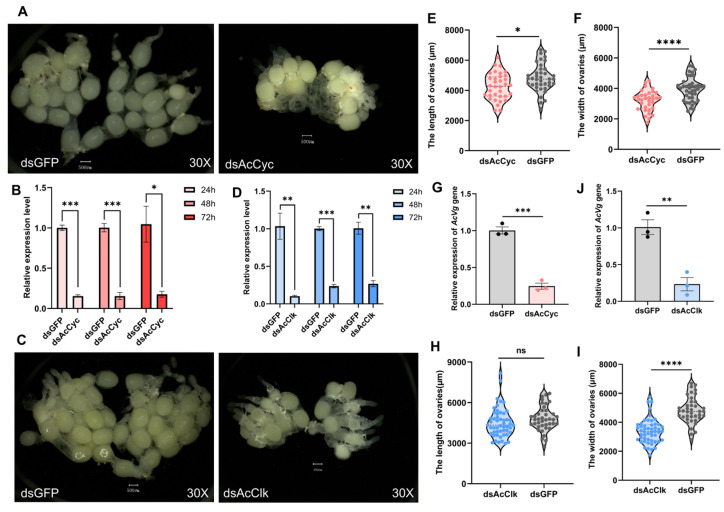
Effect of *AcCyc* or *AcClk* knockdown on ovarian development in *A. chinensis* female under non-diapause conditions. (**A**,**C**) Representative ovarian samples after dsRNA treatments. (**B**,**D**) Knockdown efficiency of *AcCyc* or *AcClk* at 24, 48 and 72 h after dsRNA treatment. (**E**,**F**,**H**,**I**) Ovary sizes (length and width) following *AcCyc* or *AcClk* silencing. (**G**,**J**) Expression of vitellogenin (Vg) after *AcCyc* or *AcClk* knockdown. Data represent mean ± stand error of mean (SEM). Asterisks indicate significant differences between dsGFP and dsRNA using Student’s *t*-test (* *p* < 0.05; ** *p* < 0.01; *** *p* < 0.001; **** *p* < 0.0001; “ns” indicate no signiffcant differences). The scale bar size was 500 μm.

**Figure 5 insects-16-01192-f005:**
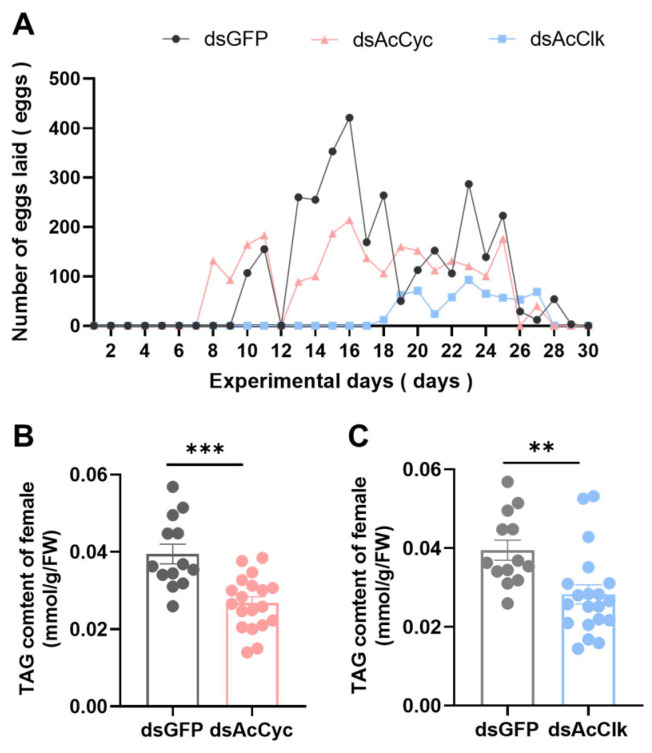
Effect of *AcCyc* or *AcClk* knockdown on reproductive capacity and energy metabolism in *A. chinensis* under non-diapause conditions. (**A**) Number of eggs laid by *A. chinensis* with *AcCyc* or *AcClk* knockdown under non-diapause conditions. (**B**,**C**) Effect of *AcCyc* or *AcClk* knockdown on TAG content in *A. chinensis* under diapause conditions. Data represent mean ± stand error of mean (SEM). Asterisks indicate significant differences between dsGFP and dsRNA using Student’s *t*-test (** *p* < 0.01; *** *p* < 0.001).

**Figure 6 insects-16-01192-f006:**
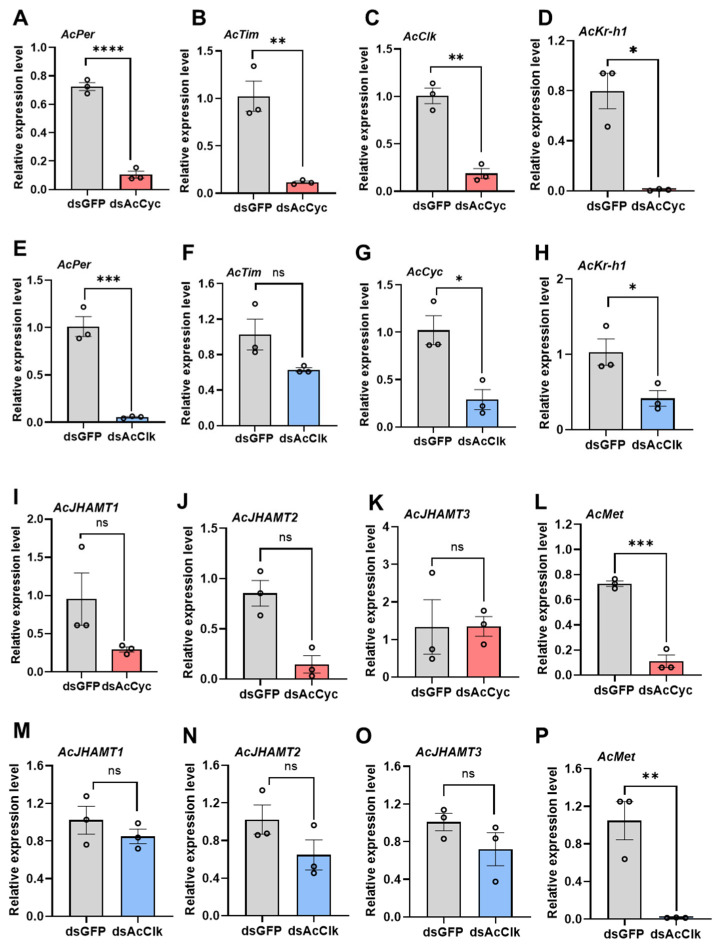
Effect of *AcCyc* or *AcClk* knockdown on juvenile hormone pathway genes and other circadian clock genes under non-diapause conditions. The expression levels of *AcPer* (**A**), *AcTim* (**B**), *AcClk* (**C**), *AcKr-h1* (**D**), *AcJHAMT1* (**I**), *AcJHAMT2* (**J**), *Ac JHAMT3* (**K**), and *AcMet* (**L**) genes after RNAi-*AcCyc*. The expression levels of *AcPer* (**E**), *AcTim* (**F**), *AcClk* (**G**), *AcKr-h1* (**H**), *AcJHAMT1* (**M**), *AcJHAMT2* (**N**), *Ac JHAMT3* (**O**), and *AcMet* (**P**) genes after RNAi-*AcClk*. Data represent mean ± stand error of mean (SEM). Asterisks indicate significant differences between dsGFP and dsRNA using Student’s *t*-test (* *p* < 0.05; ** *p* < 0.01; *** *p* < 0.001; **** *p* < 0.0001; “ns” indicate no signiffcant differences).

## Data Availability

The original contributions presented in this study are included in the article/Appendix A. Further inquiries can be directed to the corresponding author.

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
