# Peer review of "Silencing the Circadian Clock Genes Cycle and Clock Disrupts Reproductive–Metabolic Homeostasis but Does Not Induce Reproductive Diapause in Arma chinensis"

_insects, 2025, doi:10.3390/insects16121192_

Round 1
Reviewer 1 Report
Comments and Suggestions for Authors
In this manuscript, the authors present an interesting and well-executed study examining the roles of circadian clock genes Cycle (AcCyc) and Clock (AcClk) in regulating reproductive and metabolic processes in the predatory insect Arma chinensis. The authors provide substantial experimental evidence that these clock genes are crucial for reproductive and metabolic homeostasis under non-diapause conditions, and their disruption impairs ovarian development, fecundity, and triglyceride content. The use of RNA interference (RNAi) to silence these genes is a strength, and the findings suggest that AcCyc and AcClk function beyond the regulation of diapause induction, positioning them as central regulators of daily physiological coordination. The study is well-structured, with clear research objectives and solid experimental design. The results advance our understanding of circadian clock gene functions in non-diapause conditions, which is novel in the context of insect biology, especially for biological control agents like Arma chinensis.
I have the following suggestions for consideration.
(1) Figure 1. The rationale for choosing these species for phylogenetic analysis should be clarified.
(2) The manuscript reports the statistical methods (ANOVA, Tukey’s test, Student's t-test), but a more detailed explanation of how multiple comparisons were handled would be useful.
(3) The knockdown efficiency of AcCyc and AcClk is reported to be high, but it would be helpful to include a discussion on potential off-target effects and how these were minimized.
Author Response
In this manuscript, the authors present an interesting and well-executed study examining the roles of circadian clock genes Cycle (AcCyc) and Clock (AcClk) in regulating reproductive and metabolic processes in the predatory insect Arma chinensis. The authors provide substantial experimental evidence that these clock genes are crucial for reproductive and metabolic homeostasis under non-diapause conditions, and their disruption impairs ovarian development, fecundity, and triglyceride content. The use of RNA interference (RNAi) to silence these genes is a strength, and the findings suggest that AcCyc and AcClk function beyond the regulation of diapause induction, positioning them as central regulators of daily physiological coordination. The study is well-structured, with clear research objectives and solid experimental design. The results advance our understanding of circadian clock gene functions in non-diapause conditions, which is novel in the context of insect biology, especially for biological control agents like Arma chinensis.
Response to the reviews
We want to express our gratitude to you for your review of our paper and your insightful comments and suggestions which we feel have improved the quality of our manuscript. We hope the responses below address the raised issues. In addition, we have checked the full text carefully and corrected some errors on grammar and spelling that we have found to improve the quality of this manuscript.
Comment 1: Figure 1. The rationale for choosing these species for phylogenetic analysis should be clarified.
Response 1: We appreciate your comment. We have thoroughly revised our phylogenetic analysis (revised figure 1). The new phylogenetic tree included the well-established sequences of Clock and Cycle from Drosophila melanogaster and Methoprene-tolerant (Met) from Drosophila melanogaster (as an outgroup) to ensure a more balanced and representative taxon sampling. The phylogenetic analysis of AcCyc and AcClk was conducted using the Jones-Taylor-Thornton (JTT)-based neighbor-joining method with 1000 bootstrap replicates.
Comment 2: The manuscript reports the statistical methods (ANOVA, Tukey’s test, Student's t-test), but a more detailed explanation of how multiple comparisons were handled would be useful.
Response 2: Thank you for your comment. We described the statistical methods in detail in “Materials and Methods” (line 208-217), as following, the expression profiles of AcCyc and AcClk at different developmental stages or in different tissues were analyzed by one-way ANOVA, followed by a Turkey's HSD multiple comparison test with a, b & c indicating signification differences at P < 0.05. The diel expression of AcCyc and AcClk under non-diapause and diapause conditions were analyzed by one-way ANOVA, followed by a Turkey's HSD multiple comparison test, with a, b & c indicating signification differences at P < 0.05. The knockdown efficiency, TAG content, ovary sizes (length and width) and expression of genes following AcCyc or AcClk silencing were analyzed by Student’s t-test, and significance levels were denoted by * (0.01 ≤ P < 0.05), ** (0.001 ≤ P < 0.01), *** (P < 0.001) and **** (P < 0.0001).
Comment 3: The knockdown efficiency of AcCyc and AcClk is reported to be high, but it would be helpful to include a discussion on potential off-target effects and how these were minimized.
Response 3: Thank you for your comment. The potential off-targets of dsAcCyc and dsAcClk in Arma chinensis were evaluated using “dsRIP” (https://dsrip.uni-goettingen.de/efficiency) with 1 mismatche per siRNA for off-target prediction. As results, dsAcClk was designed to only specifically target AcClk gene (Cluster-9636.42629: A. chinensis Clock), and no potential off-target effects were detected, theoretically (Figure S1A; Table S3.1). The dsAcCyc was designed to specifically target AcClk gene (Cluster-9636.39048: A. chinensis Cycle), and two potential off-target genes (Cluster-9636.34927 and Cluster-9636.74299: both A. chinensis uncharacterized genes), theoretically (Figure S1B; Table S3.2).
These analyses and results have added into the manuscript, in “Materials and Methods” (lines 200-204), “Results” (lines 338-344) and “Discussion” (lines 428-431).
Reviewer 2 Report
Comments and Suggestions for Authors
The study focuses on the role of Cyc and Clock in the predatory bug Arma chinensis. Although the study is interesting, there are several issues that deserve closer attention. Firstly, I see major problems with the introduction and discussion, which neglect quite an important part of the literature. Additional issues include the phylogenetic analysis, some data presentation, and the discussion section. More specific comments are below:
Small typo in the intro, line 80.: “Helicoverpa armigera [20], Pyrrhocoris apterus [21, 22], and et al.” Clarify the end of the sentence.
Figure 1 and the corresponding phylogenetic analysis. Evolutionary relationship of Cycle and Clock. I would say that the original studies describing Clock and cycle deserve to be mentioned (doi: 10.1016/S0092-8674(00)81440-3, 10.1016/S0092-8674(00)81441-5). I do not understand why Drosophila melanogaster is missing; instead, there are up to five locust species (four of them from the genus Schistocerca!). The analysis can also be performed on both bHLH-PAS proteins together with other bHLH-PAS representatives serving as outgroups (as illustrated in doi:10.1016/j.ibmb.2025.104298). In either case, it is important to include some well-established sequences (such as Drosophila melanogaster, where Clock and Cyc were discovered) and some outgroup sequences, such as Tgo, Met, and SIM from the same species (e.g. Drosophila melanogaster, Halyomorpha halys, etc.). Please, make sure that you clearly specify the accession numbers used in the tree.
The insect taxonomy is wrong. Firstly, Hemiptera is a higher taxon than Homoptera. Furthermore, the term Homoptera is rather obsolete and not used anymore in the literature (for current taxonomy, see, for example, doi:10.1073/pnas.1815820115). Please, stick to the current taxonomy (Hemiptera contain the following groups: Heteroptera, Auchenorrhyncha, Sternorrhyncha).
Fig 2. Excellent that panels C, D, G, and H contain actual measurement values (dots). Please add these individual biological replicates also to panels A, B, E, and F. Furthermore, please use some color that contrasts better with the column background.
Fig. 3. The actual values for individual biological replicates (not only the mean and whiskers) should be shown in Figure 3.
The discussion part is very brief and needs to be extended. The topic of diapause and Clock/cyc has been extensively addressed in Heteroptera by two research groups (Shin G. Goto and David Dolezel), thus, a decent body of comparative data is available. For example, some seminal papers by Ikeno et al. deserve to be mentioned. Similarly, I do not understand why some other studies focusing on diapause and Clock/cyc are ignored (e.g., 10.1073/pnas.2510550122; or doi:10.1126/science.ado2129) and instead, a study on timeless gene is mentioned (Suri, V.; Lanjuin, A.; Rosbash, M. TIMELESS-dependent positive and negative autoregulation in the Drosophila circadian clock).
RNAi knockdowns are never perfect. Thus, one question is whether the likely residual mRNA might be responsible for the phenotype. As Figure 5 shows, there is indeed a delay in egg laying in dsAcClk, suggesting some involvement of Cyc in reproduction. This should be discussed.
Interestingly, expression of Met after Clock and Cyc RNAi is dramatically reduced. This discovery should be discussed as well. It is, for example, quite interesting that a genetic interaction among Clock, Cyc, Met and Tai was found in another Heteropteran insect, Pyrrhocoris apterus (doi: 10.1073/pnas.1217060110, doi: 10.1371/journal.pgen.1010924).
The JHAMT multiple paralogs should be discussed a bit more. Are all of them active/involved in JH synthesis? Are there similar cases where multiple JHAMT genes are found? There is a recent paper on JHAMT evolution in insects, including multiple JHAMTS in Heteroptera (e.g. doi: 10.1016/j.jinsphys.2023.104487).
Alltogether, I suggest a mojor text revision focussing much better on puttin the observed measurments to the context of published literature.
Author Response
Reviewer #2:
The study focuses on the role of Cyc and Clock in the predatory bug Arma chinensis. Although the study is interesting, there are several issues that deserve closer attention. Firstly, I see major problems with the introduction and discussion, which neglect quite an important part of the literature. Additional issues include the phylogenetic analysis, some data presentation, and the discussion section. Alltogether, I suggest a mojor text revision focussing much better on puttin the observed measurments to the context of published literature. More specific comments are below:
Comment 1: Small typo in the intro, line 80.: “Helicoverpa armigera [20], Pyrrhocoris apterus [21, 22], and et al.” Clarify the end of the sentence.
Response 1: We sincerely thank the reviewer for pointing out this oversight. The use of “and et al.” is indeed redundant. We have corrected the sentence to eliminate the repetition and improve clarity. We have revised the sentence in the introduction (line 81)
Comment 2: Figure 1 and the corresponding phylogenetic analysis. Evolutionary relationship of Cycle and Clock. I would say that the original studies describing Clock and cycle deserve to be mentioned (doi: 10.1016/S0092-8674(00)81440-3, 10.1016/S0092-8674(00)81441-5). I do not understand why Drosophila melanogaster is missing; instead, there are up to five locust species (four of them from the genus Schistocerca!). The analysis can also be performed on both bHLH-PAS proteins together with other bHLH-PAS representatives serving as outgroups (as illustrated in doi:10.1016/j.ibmb.2025.104298). In either case, it is important to include some well-established sequences (such as Drosophila melanogaster, where Clock and Cyc were discovered) and some outgroup sequences, such as Tgo, Met, and SIM from the same species (e.g. Drosophila melanogaster, Halyomorpha halys, etc.). Please, make sure that you clearly specify the accession numbers used in the tree.
Response 2: We sincerely thank the reviewer for their comments and valuable suggestions. We have thoroughly revised our phylogenetic analysis (revised figure 1). The new phylogenetic tree included the well-established sequences of Clock and Cycle from Drosophila melanogaster to ensure a more balanced and representative taxon sampling. Additionally, we used Met from Drosophila melanogaster as an outgroup. All sequences used in this revised analysis, including their accession numbers, are explicitly listed in Table S2 of the supplementary materials.
Comment 3: The insect taxonomy is wrong. Firstly, Hemiptera is a higher taxon than Homoptera. Furthermore, the term Homoptera is rather obsolete and not used anymore in the literature (for current taxonomy, see, for example, doi:10.1073/pnas.1815820115). Please, stick to the current taxonomy (Hemiptera contain the following groups: Heteroptera, Auchenorrhyncha, Sternorrhyncha).
Response 3: Thank you for your comment again. We have thoroughly revised our phylogenetic analysis sticking to the current taxonomy (revised figure 1).
Comment 4: Excellent that panels C, D, G, and H contain actual measurement values (dots). Please add these individual biological replicates also to panels A, B, E, and F. Furthermore, please use some color that contrasts better with the column background.
Response 4: Thank you for your positive feedback and valuable suggestion. We have carefully revised the figures according to your comment. Specifically, we have added individual biological replicates (dots) to panels A, B, E, and F to ensure consistency and transparency across all datasets. Additionally, we have optimized the color scheme to improve contrast with the column background, enhancing visual clarity and readability.
Comment 5: The actual values for individual biological replicates (not only the mean and whiskers) should be shown in Figure 3.
Response 5: Thank you for your comment. We have made the suggested modifications to the Figure 3.
Comment 6: The discussion part is very brief and needs to be extended. The topic of diapause and Clock/cyc has been extensively addressed in Heteroptera by two research groups (Shin G. Goto and David Dolezel), thus, a decent body of comparative data is available. For example, some seminal papers by Ikeno et al. deserve to be mentioned. Similarly, I do not understand why some other studies focusing on diapause and Clock/cyc are ignored (e.g., 10.1073/pnas.2510550122; or doi:10.1126/science.ado2129) and instead, a study on timeless gene is mentioned (Suri, V.; Lanjuin, A.; Rosbash, M. TIMELESS-dependent positive and negative autoregulation in the Drosophila circadian clock).
Response 6: Thank you for your comment. We have expanded the “Discussion” in our manuscript as follows (lines 368-377): Our findings provide compelling evidence that in the predatory A. chinensis, AcClk and AcCyc function as indispensable regulators of reproductive-metabolic homeostasis under favorable (non-diapause) conditions. This aligns with observations in other hemipteran species. For instance, in the brown-winged green stink bug Plautia stali, RNAi-mediated knockdown of Cyc disrupts the photoperiodic response, leading to suppressed ovarian development [33]. Specifically, Cyc RNAi resulted in inhibited ovarian development in 45% of individuals under long-day conditions, a phenotypic outcome consistent with the effects observed following AcCyc knockdown in our study. These collective results suggest a conserved role for the Cycle in Hemiptera, underscoring its core function in the daily coordination of reproduction.
Reference
[33] Tamai, T.; Shiga, S.; Goto, S. G. Roles of the circadian clock and endocrine regulator in the photoperiodic response of the brown-winged green bug Plautia stali. Physiological Entomology. 2019, 44, 43-52.
Comment 7: RNAi knockdowns are never perfect. Thus, one question is whether the likely residual mRNA might be responsible for the phenotype. As Figure 5 shows, there is indeed a delay in egg laying in dsAcClk, suggesting some involvement of Cyc in reproduction. This should be discussed.
Response 7: Thank you for your comment. Although we don't fully understand your meaning, I guess your question is “whether the likely residual mRNA might be responsible for the phenotype. As Figure 5 shows, there is indeed a delay in egg laying in dsAcClk, suggesting some involvement of Clk in reproduction”.
We sincerely thank the reviewer for an important point regarding the interpretation of RNAi knockdown experiments and the potential contribution of residual mRNA to the observed phenotypic outcomes. We have expanded the “Discussion” in our manuscript as follows (lines 419-431):
The absence of a complete and immediate diapause phenotype following RNAi could be attributed to several factors: (1) functional redundancy or compensatory mechanisms within the circadian clock network, where other genes buffer the loss of a single component; (2) the action of downstream outputs that remain partially active even when the core clock is dampened; or (3) species-specific differences in the reliance on clock genes for diapause induction. Furthermore, RNAi knockdowns are inherently imperfect. The residual mRNA and protein activity might be sufficient to prevent the full manifestation of the phenotype, a possibility underscored by the delayed egg-laying observed in dsAcClk-treated insects (Figure 5), which suggests a partial rather than absolute loss of function. While our RNAi efficiencies were high for both dsAcCyc and dsAcClk (Figure 4B, D), we acknowledge the inherent limitations of the technique, including the potential for non-specific off-target effects (theoretically predicted for dsAcCyc in Figure S1B and Table S3.2). The RNAi approach, while powerful, can lead to non-specific effects and typically results in partial rather than complete loss-of-function phenotypes, potentially underestimating a gene's role if compensatory mechanisms exist. Future investigations employing more precise gene-editing technologies like CRISPR/Cas9, or combinatorial gene knockdowns, would provide deeper insights into the functional hierarchy within the clock and its outputs.
Comment 8: Interestingly, expression of Met after Clock and Cyc RNAi is dramatically reduced. This discovery should be discussed as well. It is, for example, quite interesting that a genetic interaction among Clock, Cyc, Met and Tai was found in another Heteropteran insect, Pyrrhocoris apterus (doi: 10.1073/pnas.1217060110, doi: 10.1371/journal.pgen.1010924).
Response 8: We thank the reviewer for highlighting the importance of our finding that AcCyc and AcClk knockdown dramatically reduced Met expression and for directing us to the highly relevant and important studies in Pyrrhocoris apterus. As suggested, we have now incorporated a discussion of this finding and the cited references into the revised “Discussion” in manuscript (lines 383-389). As following, the dramatic reduction in Met expression following AcCyc and AcClk knockdown implied genetic interactions between circadian clock components and JH signaling pathways. This genetic interaction between circadian clock components and JH signaling elements (eg, Met, Taiman), also previous reported in photoperiodic regulation of P. apterus [34, 35], appears to be a conserved regulatory module in Hemiptera insects.
References
[34] Bajgar, A.; Jindra, M.; Doleze, David. Autonomous regulation of the insect gut by circadian genes acting downstream of juvenile hormone signaling PNAS. 2013,110 (11): 4416-4421
[35] Smykal, V.; Chodakova, L.; Hejnikova, M.; Briedikova, K.; Wu, B. C-H.; Vaneckova, H., Chen, P.; Janovska, A.; Kyjakova, P.; Vacha, M.; Dolezel D. Steroid receptor coactivator TAIMAN is a new modulator of insect circadian clock. PLoS Genet. 2023, 19(9): e1010924.
Comment 9: The JHAMT multiple paralogs should be discussed a bit more. Are all of them active/involved in JH synthesis? Are there similar cases where multiple JHAMT genes are found? There is a recent paper on JHAMT evolution in insects, including multiple JHAMTS in Heteroptera (e.g. doi: 10.1016/j.jinsphys.2023.104487).
Response 9: We sincerely thank the reviewer for this insightful comment. we have expanded more detailed discussion in our revised manuscript (lines 391-406). As following, However, the expression levels of three JHAMT genes, AcJHAMT1, AcJHAMT2, and AcJHAMT3, remained unchanged following RNAi. The copy number of JHAMT genes exhibits considerable variation across arthropod species. While some insects, such as Drosophila melanogaster, Apis mellifera, and Aphis craccivora, possess only a single JHAMT gene in their genomes [40], the occurrence of multiple JHAMT paralogs is remarkably widespread in insects, including in the desert locust Schistocerca gregaria (31 JHAMT-like genes), the housefly Musca domestica (12 JHAMT genes), and the silkworm Bombyx mori (6 JHAMT genes) [40]. However, the presence of multiple JHAMT gene copies does not imply that all are functionally involved in JH biosynthesis. For example, in the red flour beetle, Tribolium castaneum, which possesses three JHAMT genes, only one encodes an enzyme with genuine JH acid methyltransferase activity, while the other two paralogs lack this catalytic function [41]. It is important to note that, although direct evidence is currently lacking to confirm that all three AcJHAMT genes are functionally involved in JH synthesis in A. chinensis, the absence of significant changes in their transcript levels following RNAi-AcCyc/AcClk may suggested that the transcriptional regulation of JH biosynthesis was not substantially impacted after RNAi-AcCyc/AcClk.
Reference
[40] Smykal, V.; Dolezel, D. Evolution of proteins involved in the final steps of juvenile hormone synthesis. Journal of Insect Physiology. 2023, 145: 104487
[41] Minakuchi, C.; Namiki, T.; Yoshiyama, M.; Shinoda, T. RNAi-mediated knockdown of juvenile hormone acid O-methyltransferase gene causes precocious metamorphosis in the red flour beetle Tribolium castaneum. FEBS J. 2008, 275 (11), 2919-2931.
Reviewer 3 Report
Comments and Suggestions for Authors
In the current study, the authors investigate the roles of core circadian clock genes Cycle (AcCyc) and Clock (AcClk) in Arma chinensis, a biological control agent. They characterize these genes and analyze their expression patterns, using RNA interference to knockdown AcCyc and AcClk in non-diapausing females. The results reveal that knockdown significantly impairs reproduction, decreasing ovarian size, vitellogenin expression, egg production, and triglyceride levels, indicating disrupted energy homeostasis. The study concludes that AcCyc and AcClk are crucial for maintaining reproductive and metabolic balance beyond their known role in diapause. Overall, the study is interesting. The manuscript is well-written, and the data are presented in a clear manner. I can only think of a few small edits for improvement.
1- Please explain all abbreviations when they are first mentioned in text. For example, this is not done for (Met, Kr-h1).
2- At line 117, it is stated that samples included female adults at 0, 3, 6, 9, and 12 days, but in figs. 2A and 2E, I see that it was also sampled at 15 days. Please verify.
3- It is not clear on what basis the sampling days (lines 117-118) were identified under diapause and non-diapause conditions.
4- The statement at lines 122-123 should be removed since this is not the same for different sets of experiments at lines 155, 171, 181, and 192.
5- At line 166, it is unclear what you mean by “2 μg” of dsRNA was injected. Do you mean microliter (μl) form of the dsRNA solution? It is stated that 1 μl of dsRNA solution is used at line 189.
6- At lines 201-202: There appears to be a discrepancy between the asterisks and those represented in figures and legends (e.g., lines 292-293), where ****P<0.0001 is also included.
7- Figure 1, Page 6: Please specify that panel “A” is for Cycle and panel “B” is for Clock proteins at line 215. Also, in Fig. 1B, the “Ranatra chinensis” should be italicized.
8- In Fig. 2, page 7, it is a bit confusing to use abbreviations like N, D, and NE without defining these in the legends, as some of them are already used for other purposes in similar studies. Please define.
9- At line 242, it is stated that the AcCyc was high at ZT6, but I also see another peak at ZT18.
10- For Fig. 4A, there is a typo; it should be “dsGFP” instead of “dsGF”. Also, I would like to mention the size of the scale bar in the legend text, as it is not clearly visible in Figs. 4A and 4B.
11- At line 303, Figure 5, it is stated that “AcPer or AcTim knockdown”; if I am not mistaken, this should be AcCyc and AcClk as described in lines 279-285. Also, at line 285, it should be (Fig. 5C) for AcClk silencing, and (Fig. 5B) should be moved earlier in the same line for AcCyc.
12- Please ensure that the format of the gene names is consistent throughout the manuscript and matches that of similar studies. For example, at lines 308 and 313, “AcMet” has only the first letter capitalized (i.e., Met). However, this is not the case at lines 318 and 320, where they are capitalized (i.e., AcMET), and in Figs. 6L and 6P.
13- In Fig. 6, there is not enough explanation in the text or figure legend about the time of the injection, ages, or sampling.
14- Figure-related: The colors of the different lines and bars representing the treatments are inconsistent between figures. It is confusing, and I would suggest, when possible, to alter your color scheme to have the same color for each treatment (AcCyc or AcClk knockdown) and use grey for the control (GFP) to make it easier to follow.
15- Please review the appearance of “T7: GATCACTAATACGACTCACTATAGGG” beneath Table S1.
Author Response
Reviewer #3:
In the current study, the authors investigate the roles of core circadian clock genes Cycle (AcCyc) and Clock (AcClk) in Arma chinensis, a biological control agent. They characterize these genes and analyze their expression patterns, using RNA interference to knockdown AcCyc and AcClk in non-diapausing females. The results reveal that knockdown significantly impairs reproduction, decreasing ovarian size, vitellogenin expression, egg production, and triglyceride levels, indicating disrupted energy homeostasis. The study concludes that AcCyc and AcClk are crucial for maintaining reproductive and metabolic balance beyond their known role in diapause. Overall, the study is interesting. The manuscript is well-written, and the data are presented in a clear manner. I can only think of a few small edits for improvement.
Comment 1: Please explain all abbreviations when they are first mentioned in text. For example, this is not done for (Met, Kr-h1).
Response 1: We appreciate the reviewer's attention to this important detail. All abbreviations throughout the manuscript have explained when they are first mentioned. eg Methoprene-tolerant (Met) and, Krueppel homolog 1 (Kr-h1) and Juvenile hormone acid methyltransferase enzyme (JHAMT).
Comment 2: At line 117, it is stated that samples included female adults at 0, 3, 6, 9, and 12 days, but in figs. 2A and 2E, I see that it was also sampled at 15 days. Please verify.
Response 2: Thank you for your comment. We have verified that the samples included female adults at 0, 3, 6, 9, 12 and 15 days under non-diapause conditions. We have revised it in line 119.
Comment 3: It is not clear on what basis the sampling days (lines 117-118) were identified under diapause and non-diapause conditions.
Response 3: Thank you for your comment. We are pleased to provide a detailed explanation below: The selection of sampling days was based on preliminary data from our laboratory (unpublished) characterizing the physiological progression of diapause and reproduction in A. chinensis. Adult females enter a well-defined three diapause process when transferred to diapause-inducing conditions (short-day photoperiod and low temperature): (1) Diapause induction preparation phase (Days 1-40 under diapause conditions); (2) Diapause maintenance phase (Days 40-120); (3) post-diapause phase (beyond Day 120). Accordingly, we selected time points within the induction and early maintenance phases (0, 10, 20, 30, 40, and 50 days) to capture dynamic changes in clock gene expression during critical transitions. Under non-diapause (reproductively active) conditions, females typically initiate oviposition between 8–12 days after emergence. To monitor gene expression patterns throughout pre-oviposition and early reproductive stages, we sampled at shorter intervals (0, 3, 6, 9, 12, and 15 days).
Comment 4: The statement at lines 122-123 should be removed since this is not the same for different sets of experiments at lines 155, 171, 181, and 192.
Response 4: Thank you for your comment. We have removed the statement at lines 122-123.
Comment 5: At line 166, it is unclear what you mean by “2 μg” of dsRNA was injected. Do you mean microliter (μl) form of the dsRNA solution? It is stated that 1 μl of dsRNA solution is used at line 189.
Response 5: Thank you for your comment. The concentration of dsRNA was 2 µg/μL, and the injection volume was 1 μL. We have provided a more accurate description of the usage of dsRNA in manuscript (lines 170)
Comment 6: At lines 201-202: There appears to be a discrepancy between the asterisks and those represented in figures and legends (e.g., lines 292-293), where ****P<0.0001 is also included.
Response 6: Thank you for your comment. We have revised the “Statistical analysis” in manuscript and added “****(P<0.0001)” in line 217
Comment 7: Figure 1, Page 6: Please specify that panel “A” is for Cycle and panel “B” is for Clock proteins at line 215. Also, in Fig. 1B, the “Ranatra chinensis” should be italicized.
Response 7: Thank you for your comment. We have thoroughly revised our phylogenetic analysis (revised figure 1). All Latin names have been italicized.
Comment 8: In Fig. 2, page 7, it is a bit confusing to use abbreviations like N, D, and NE without defining these in the legends, as some of them are already used for other purposes in similar studies. Please define.
Response 8: Thank you for your comment. We have now added the following explanatory sentence to the legend of Figure 2 (lines 251-253). As following, NE, N3, N6, N9, N12, and N15 represent female adults at 0, 3, 6, 9, 12, and 15 days post-eclosion under non-diapause conditions, respectively; NE, D10, D20, D30, D40, and D50 represent female adults at 0, 10, 20, 30, 40, and 50 days post-eclosion under diapause conditions, respectively.
Comment 9: At line 242, it is stated that the AcCyc was high at ZT6, but I also see another peak at ZT18.
Response 9: Thank you for your comment. We have corrected it in manuscript (line 258).
Comment 10: For Fig. 4A, there is a typo; it should be “dsGFP” instead of “dsGF”. Also, I would like to mention the size of the scale bar in the legend text, as it is not clearly visible in Figs. 4A and 4B.
Response 10: Thank you for your comment. We sincerely thank the reviewer for their keen observation. The typo in Fig. 4A has been corrected to "dsGFP" as suggested. Additionally, we have now clearly stated the scale bar size (500 μm) in the figure 4 legend (line 310).
Comment 11: At line 303, Figure 5, it is stated that “AcPer or AcTim knockdown”; if I am not mistaken, this should be AcCyc and AcClk as described in lines 279-285. Also, at line 285, it should be (Fig. 5C) for AcClk silencing, and (Fig. 5B) should be moved earlier in the same line for AcCyc.
Response 11: We sincerely thank the reviewer for their meticulous reading and for identifying these inconsistencies in the text descriptions and figure citations. We have carefully revised the manuscript to address these errors have ensured that all gene names and figure citations are now accurate and consistent throughout the text.
Comment 12: Please ensure that the format of the gene names is consistent throughout the manuscript and matches that of similar studies. For example, at lines 308 and 313, “AcMet” has only the first letter capitalized (i.e., Met). However, this is not the case at lines 318 and 320, where they are capitalized (i.e., AcMET), and in Figs. 6L and 6P.
Response 12: We thank the reviewer for their careful reading and for pointing out the inconsistency in gene nomenclature. We have now carefully reviewed the entire manuscript and standardized the formatting of all gene names to ensure consistency. Specifically, we have corrected "AcMET" to "AcMet" in the text (lines 330, 334, and elsewhere) and in Figures 6L and 6P, so that the gene symbol now consistently appears as "AcMet" throughout the manuscript.
Comment 13: In Fig. 6, there is not enough explanation in the text or figure legend about the time of the injection, ages, or sampling.
Response 13: Thank you for your comment. We have added the explanation in the “Materials and Methods” (lines 172-175). Female adults (24 h after emergence) was injected with dsRNA under non-diapause conditions. The expressions of AcPer gene, AcTim gene, and juvenile hormone pathway genes (AcJHAMT1, AcJHAMT2, AcJHAMT3, AcKr-h1 and AcMet) were assessed at 48 h post-injection.
Comment 14: Figure-related: The colors of the different lines and bars representing the treatments are inconsistent between figures. It is confusing, and I would suggest, when possible, to alter your color scheme to have the same color for each treatment (AcCyc or AcClk knockdown) and use grey for the control (GFP) to make it easier to follow.
Response 14: We thank the reviewer for this valuable suggestion to improve the consistency and clarity of our figures. As suggested, we have now revised the color scheme in all relevant figures. Throughout the manuscript, the control group (dsGFP) is now consistently represented in grey. The treatment groups (e.g., dsAcCyc and dsAcClk knockdown) are each represented by a distinct color that is kept consistent across all figures.
Comment 15: Please review the appearance of “T7: GATCACTAATACGACTCACTATAGGG” beneath Table S1.
Response 15: Thank you for your comment. We have revised the footnote to Table S1 to more clearly and formally describe the purpose of the T7 sequence. The note now explicitly states that the T7 sequence is the promoter sequence used for in vitro transcription of dsRNAs.
Round 2
Reviewer 2 Report
Comments and Suggestions for Authors
I appreciate the correction of the figures and the extended discussion. Since the paper focuses on CLOCK/CYCLE and its (lack of) role in diapause, I also suggest mentioning two additional studies. For example, the paper with DOI: 10.1073/pnas.2510550122 (Noncanonical action of circadian clock genes controls winter diapause entry via the NuA4/TIP60 complex in Harmonia axyridis) provides an example of diapause regulation where only some circadian clock genes play a role, whereas others do not. Such dissociation within the mechanism could also be relevant to this study.
Author Response
Response to the reviewer#2 (Round 2)
Reviewer #2: I appreciate the correction of the figures and the extended discussion. Since the paper focuses on CLOCK/CYCLE and its (lack of) role in diapause, I also suggest mentioning two additional studies. For example, the paper with DOI: 10.1073/pnas.2510550122 (Noncanonical action of circadian clock genes controls winter diapause entry via the NuA4/TIP60 complex in Harmonia axyridis) provides an example of diapause regulation where only some circadian clock genes play a role, whereas others do not. Such dissociation within the mechanism could also be relevant to this study.
Response: We sincerely thank the reviewer for this additional insightful suggestion. In response to the reviewer's comment, we have expanded the discussion in our manuscript to include a comparative analysis of two study (https://doi.org/10.1073/pnas.2510550122 and doi:10.1126/science.ado2129). The added discussion paragraphs are as follows (inserted into Section 4. Discussion of the manuscript) (lines 409-433):
Furthermore, recent research in Lepidoptera has revealed that the circadian clock gene Cycle exhibits functional polymorphism through alternative isoforms, which play distinct roles in diapause regulation [42]. In the silk moth Bombyx mori, Zheng et al. (2025) demonstrated that Cyc encodes three major isoforms: CycA and CycB are primarily involved in core circadian rhythm maintenance, while CycC specifically regulates diapause entry [42]. A critical deletion in the CycC isoform disrupts diapause induction in polyvoltine strains, leading to nondiapause phenotypes, without affecting circadian rhythms controlled by CycA/B. This isoform-specific function is conserved across Lepidoptera, including distantly related species like the Asian corn borer (Ostrinia furnacalis), where CycC knockdown reduces diapause incidence [42]. These findings provide a nuanced perspective on our results in A. chinensis, where knockdown of AcCyc under non-diapause conditions impaired reproductive-metabolic homeostasis (e.g., reduced ovarian development, JH signaling disruption) but did not induce typical diapause. This suggests that in A. chinensis, the Cyc ortholog may function analogously to the rhythm-regulating isoforms (CycA/B) rather than the diapause-specific CycC, potentially due to evolutionary divergence in Hemiptera. Notably, our results different form recent findings in other Coleoptera [43]. Gao et al. (2025) demonstrated that in the ladybird beetle Harmonia axyridis, Clk and Cyc regulate winter diapause entry through a noncanonical pathway involving the NuA4/TIP60 histone acetyltransferase complex [43]. In H. axyridis, knockdown of Clk and Cyc under long-day conditions induced diapause-like phenotypes via disruption of juvenile hormone (JH) biosynthesis, whereas knockdown of Per or Tim had no effect, a dissociation similar to what we observe in A. chinensis. This comparison underscores that circadian clock genes can function through rhythm-independent mechanisms across insects, but their specific outcomes (e.g., diapause induction vs. homeostasis maintenance) may be species- or context-dependent.
References
[42] Zheng, S.R; Wang, Y. H.; Li, G. Y.; Qin, S.; Dong, Z.; Yang, X.; Xu, X. M.; Fang, G. Q.; Li, M.W.; Zhan, S. Functional polymorphism of CYCLE underlies the diapause variation in moths. 2025, 388(6750): 2129.
[]43 Gao, Q.; Dai, Y.F.; Zhao, Y.L.; Li, X.; An, H. M.; Jones, K. K.; Wang, J. L.; Wanga, X. P.; Liu, W. Noncanonical action of circadian clock genes controls winter diapause entry via the NuA4/TIP60 complex in Harmonia axyridis. Proceedings of the National Academy of Sciences of the United States of America. 2025, 122 (28): e2510550122.